# Invasibility of Common Plant Community Types of the Middle Urals

Denis V. Veselkin, Natalya V. Zolotareva, Denis I. Dubrovin *, Elena N. Podgaevskaya, Liliya A. Pustovalova and Anastasia A. Korzhinevskaya

Institute of Plant and Animal Ecology, Ural Branch of the Russian Academy of Sciences, 8 Marta St., 202, 620144 Ekaterinburg, Russia; veselkin_dv@ipae.uran.ru (D.V.V.); enp@ipae.uran.ru (E.N.P.); melnikova_aa@ipae.uran.ru (A.A.K.)
* Correspondence: dubrovin_di@ipae.uran.ru

**Abstract:** This paper specifies the invasibility of common plant community types in the natural habitats of the Middle Urals. Invasibility was defined as the vulnerability of a community to alien plant species invasions, regardless of the conditions in which the community existed. We analyzed 749 vegetation relevés made in natural bogs, floodplains, rocky grasslands, meadows, and forest communities. We surveyed urban and non-urban habitats (30–40 km from the city). Invasibility was calculated in two different ways based on two parameters: the number and proportion of alien species in the relevé. These invasibility parameters are widely applicable and comparable, scale-independent, measurable, and reliable, based on data that do not require the destruction of ecosystems or biota and are based on available data. Based on the invasibility degree, we identified three community-type groups with two subgroups in one group. In ascending order of invasibility degree, the community types were arranged as follows: (1) bogs with and without trees; (2a) coniferous forests; grass communities in floodplains; (2b) deciduous forests, coniferous plantations, floodplain communities with woody plants; and (3) dry meadows and rocky grasslands. Obtained results of the assessment of different plant community invasibility may be used to understand patterns of alien plant distribution in local habitats and the reasons for the different vulnerability of communities to plant invasions.

**Keywords:** plant invasions; invasibility; alien plants; urbanization; EUNIS



## 1. Introduction

Invasibility is the vulnerability of a community or habitat to alien species invasions [1–4]. This is now the accepted meaning of the term [1,2,4]. The community invasibility concept emerged as complementary to the alien species invasiveness concept (invasive potential). Invasibility is a complex property that can be attributed to communities and ecosystems [4] as well as habitats [5,6]. The recipient ecosystem properties potentially influencing its invasibility are diverse and include (a) resource availability and their temporal fluctuations; frequency, type, and degree of disturbance and environmental heterogeneity; (b) diversity and species richness of native plant communities associated with biotic resistance to invasions or their functional or evolutionary similarity to invasive alien species; and the presence (or absence) of natural enemies and mutualists [2,4]. The accumulated information about specific mechanisms responsible for community invasibility demonstrates that some mechanisms, such as disturbances, seem to be universal, while others, such as those associated with native species diversity, are not always confirmed and remain debatable [2,4].

There are different approaches to estimating the community invasibility degree [2]. When using continuous parameters of alien plant presence, there are at least two different approaches.

According to the first approach, the number [2,7–10] or proportion [2,5,6,9,10] of alien species in a particular relevé, habitat, or community is used to estimate its invasibility.

However, it is difficult to interpret the invasibility estimation results unambiguously. The observed number of alien species in a community is determined both by the properties of the community or habitat (i.e., invasibility itself) and by the properties of the surrounding landscape or neighboring communities. For example, even vulnerable communities may not be susceptible to invasion if there are no alien plants.

Therefore, according to the second approach, the "number/proportion of alien species" refers to the "invasion level" or "level of invasion" [5,6,8,11–13]. Invasibility is defined as potential alien plant invasion of a habitat or community type under equal environmental conditions (urbanization, distance from settlements and alien plant propagule distribution sites, disturbance, etc.) [3]. Therefore, "invasibility levels" are mathematically calculated in a more complex way than invasion levels [2,3,5,14–17]. For example, invasibility can be determined by estimating not only the number (proportion) but also the abundance or occurrence of alien species [2,14–17].

The importance of distinguishing "invasion levels" and "invasibility" is strongly emphasized in a large contemporary review of the reasons for successful plant invasions [4]. It states that "invasibility can be estimated as the probability of successful rooting of each alien propagule introduced to an ecosystem" [4]. Thus, the main difference between "invasion level" and "invasibility" is the dependence of the former on propagule pressure and the absence (practical or at least theoretical) of such a dependence on the latter.

There are many ways to explore community invasibility. First, one can formulate a specific hypothesis about the properties of invasible communities or ecosystems and test it. There are many examples of such studies [18–20]. Second, one can estimate the invasibility of many heterogeneous communities and then explain the empirical patterns obtained. There are also many examples of this approach [5,11,21,22]. In our work, we followed the second approach. The paper's objective was to compare the invasibility degree of the common plant community types in the natural habitats of the Middle Urals based on the relevés made for two urbanization levels: within a large city and in its non-urban areas. Before starting the study, we did not have any clear hypotheses about the least invasible communities. However, we assumed that floodplain communities (regularly spontaneously disturbed during spring floods), meadows (because there a no species with strong environment-creating capacity), or artificial tree plantations (because such communities have been severely disturbed as a result of cutting or fires) may be the most vulnerable to invasions.

## 2. Materials and Methods

### 2.1. Study Area

Ekaterinburg is a large city with an area of about 1.1 thousand km$^2$ [23], the fourth largest city in Russia with 1.5 million residents [24], located on the border of Europe and Asia (56°51′ N, 60°36′ E) far from seas and oceans. The city is located in the eastern foothills of the Middle Urals. The relief is hilly-ridged with heights of 250–300 m. The climate is humid continental, with an average temperature of −12.6 °C in January and +19.0 °C in July. The average annual precipitation is 450–500 mm, with the largest amount falling in the warm part of the year—300–350 mm. Snow cover sets in early November. Its average height is 40–50 cm. Sharp temperature fluctuations and weather anomalies are common for the region: thaws in winter and frosts in summer. By the Köppen–Geiger climate classification [25] climate of the region is cold (continental) without dry season, warm summer (Dfb). According to the botanical and geographical zoning, Ekaterinburg is in the southern taiga subzone. Pine forests on soddy-podzolic and brown soils are widespread in the study area. The tree layer is dominated by *Pinus sylvestris* L. with the presence of *Betula pendula* Roth. The herb layer is dominated by *Calamagrostis arundinacea* (L.) Roth, *Brachypodium pinnatum* (L.) P.Beauv., *Rubus saxatilis* L., and *Vaccinium myrtillus* L. These forests belong to the class *Brachypodio pinnati-Betuletea pendulae* Ermakov, Koroljuk et Latchinsky 1991.

Ekaterinburg is heavily polluted due to many industrial enterprises and high traffic density [26–28], and the city's vegetation is intensively used for recreational purposes [29]. Highly transformed central and less transformed peripheral regions are clearly distinguished. Industrial and residential districts have almost no natural vegetation. Forest parks and urban forests are less transformed areas, mainly represented by pine forests.

Ekaterinburg is the center of a large urban agglomeration with an area of about 13 thousand km$^2$ and a population of about 2.3 million residents, which, apart from Ekaterinburg, includes its satellite cities of Verkhnaya Pyshma, Berezovsky, Degtyarsk, Pervouralsk, Revda, Sredneuralsk, Aramil, Sysert, Polevskoy, and other settlements. This causes a significant anthropogenic transformation of landscapes, habitats, and communities in the non-urban area.

The city flora (within Ekaterinburg) is well-studied [30] and includes many alien species, but their distribution among habitat groups is mainly represented at the level of coenofloras or partial floras, i.e., at the level of γ-diversity [31,32]. Invasion levels have been previously determined for a limited range of habitat types: forest parks [33,34] and plantation fragments with native and alien tree species [35].

### 2.2. Plant Communities

Invasion levels were determined in relation to the observed common plant community types. To allow the integration of our data with the data of other authors, we used habitat classification [36] as the basis for selecting and determining the latitude of the community categories used, realizing that for the territories in the Asian part of Eurasia, this approach is conditional.

We studied plant communities of the following 10 types:

- Bog communities without trees—sedge and reedbeds fens and bogs without trees (EUNIS Q1, Q5);
- Bog communities with trees—swamps and boreal bog woodlands, shrub fens (EUNIS T16, T3J, S92);
- Floodplain communities without trees—floodplain meadows (EUNIS R35);
- Floodplain communities with trees—riparian forests formed by species of the genus *Salix*, *Padus*, *Alnus* (EUNIS T11);
- Meadows—mesic and wet meadows (EUNIS R2, R3);
- Birch forests—forests formed by species of the genus *Betula* (EUNIS T1*);*
- Coniferous plantations, generally Scots pine (*Pinus sylvestris*) (EUNIS T359);
- Coniferous forests (*Pinus sylvestris*) (EUNIS T3);
- Rocky grassland—rocky steppes of *Festuco-Brometea* class (EUNIS R1A) and anthropogenic herbaceous vegetation near the rocks (EUNIS V3);
- Forests dominated by Scots pine (*Pinus sylvestris*) on rocky outcrops (EUNIS T35).

Communities of each type were studied at two urbanization levels: within Ekaterinburg and in non-urban areas located 30–40 km from the city (Figure 1). Non-urban habitats were considered if they were located: (I) outside the official border of the city of Ekaterinburg or (II) a further 1.5 km from the nearest built-up multi-story buildings or industrial zones inside the official border of the city of Ekaterinburg.

### 2.3. Vegetation Relevés

In total, we analyzed 749 vegetation relevés (Table 1). The number of relevés differed by more than an order of magnitude between different community types, but the number of urban and non-urban relevés for each community type was close. Places for performing relevés were chosen both on the edges of habitats and inside them, i.e., far from roads, buildings, and other natural or artificial edges. Relevés were performed from 2013–2022. When we chose the sample plot sizes, we based ourselves on a "minimal area". "Minimal area" usually increases with the increase of size and spacing of the dominant plant life forms under consideration [37,38]. Forest relevés are often chosen to be one-tenth acre in size, i.e., 400 m$^2$, shrubland relevés 200 m$^2$, grassland relevés 100 m$^2$ [38]. Thus, we used relevés

of 400 m² for communities dominated by trees or shrubs (bogs with and trees, coniferous forests, deciduous forests, coniferous plantations, and woody floodplain communities) and 100 m² in herbal communities (bogs without trees, herbal floodplains, dry meadows, and rocky grasslands). Relevés of 400 m² were performed in plots 20 m × 20 m or in circular plots with a radius of 11.3 m. There were 236 circular plots situated in coniferous forests only. Relevés of 100 m² were always performed in square plots 10 m × 10 m. The relevés were performed during the period from the second half of June to the first half of August. When making the relevés, we recorded all vascular plant species found in an area of 100 or 400 m². Species that had been clearly identified in the field were recorded. Species that had not been identified in the field were collected and identified in the laboratory.

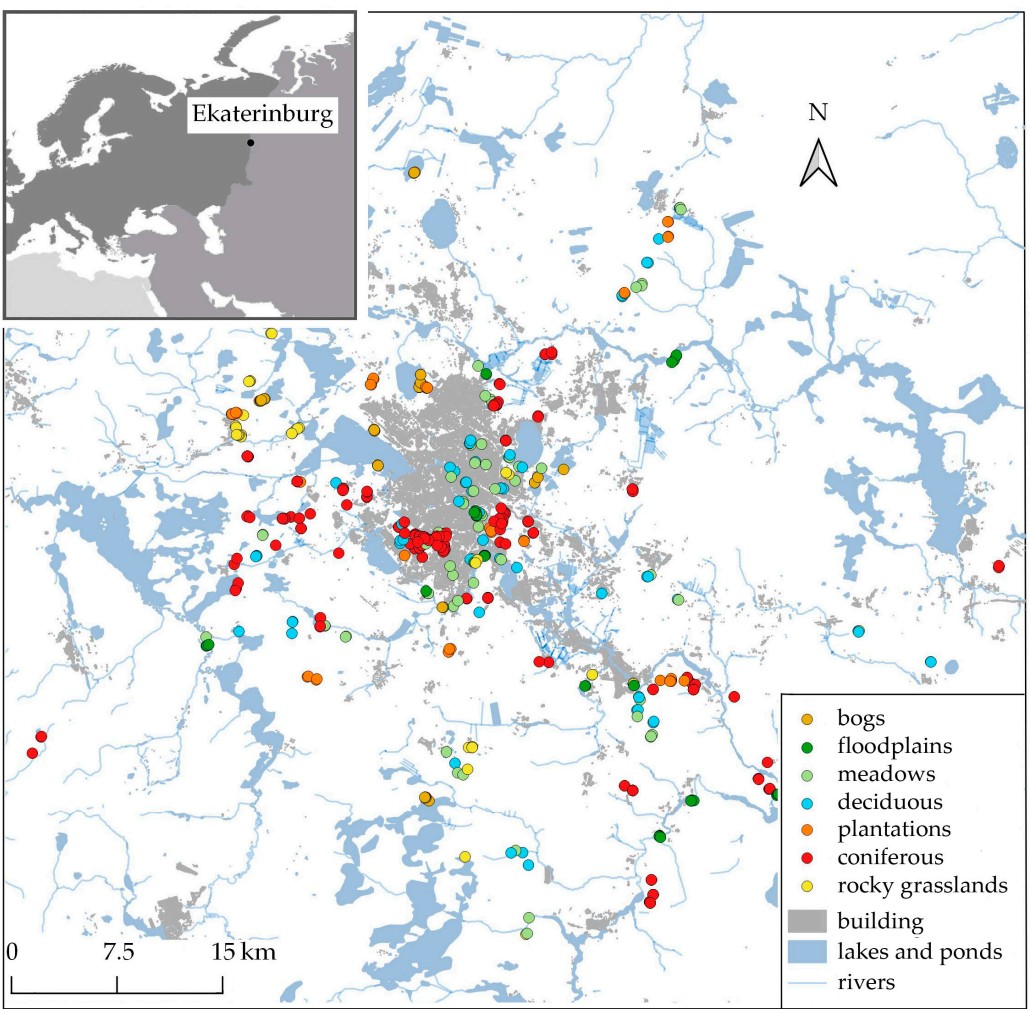

**Figure 1.** Study area map with marked vegetation relevés of different habitats. Each dot marks one or several closely spaced relevés.

### 2.4. Alien Species

All species were classified into one of two categories—native or alien. Alien species are plant taxa in a given area whose presence there is due to intentional or accidental introduction as a result of human activity (including casual alien and naturalized plants). All alien species in our study are more than 100 km away from their native populations [39]. The majority of alien species of the Middle Urals by origin belong to Mediterranean and Iran-Turanian groups, and fewer species to American and European groups [30]. Alien species to Central Europe have been traditionally divided into archaeophytes (introduced before the discovery of America in 1492) and neophytes (introduced after that date) [40]. The boundary between the archaeophytes and neophytes in the Middle Urals was determined

by 1800—the time when the Russian population engaged in agriculture appeared on this territory [30]. In this study, alien plants were not divided into archaeophytes and neophytes. The main source used to distinguish between native and non-native species was the Sverdlovsk Region Flora Abstract [41–48].

**Table 1.** Number of relevés in different community types.

| Community Types (EUNIS Habitat Types) | Urbanization Levels | | Total |
|---|---|---|---|
| | Non-Urban Habitats | Urban Habitats | |
| Fens and bogs without trees (Q1, Q5) | 14 | 25 | 39 |
| Swamps, bog woodlands, shrub fens (T16, T3J, S92) | 16 | 3 | 19 |
| Flood meadows (R35) | 20 | 15 | 35 |
| Riparian forests (T11) | 15 | 16 | 31 |
| Mesic and wet meadows (R2, R3) | 30 | 33 | 63 |
| Birch forests (T1) | 30 | 28 | 58 |
| Coniferous plantations (T359) | 26 | 26 | 52 |
| Coniferous forests (T3) | 180 | 230 | 410 |
| Rocky grasslands (R1A, V3) | 13 | 4 | 17 |
| Pine forests on rocky outcrops (T35) | 23 | 2 | 25 |
| Total | 367 | 382 | 749 |

*2.5. Invasibility Degree Estimation*

The invasibility degree was calculated based on the number of alien species ($N_{alien}$) and the proportion of alien species in the total number of species ($P_{alien}$) in the relevé. Determining community invasibility required either (1) including or (2) excluding the influence of the environment, urbanization in our case, on the $N_{alien}$ and $P_{alien}$ values. Therefore, we used two methods for calculating the invasibility. The first of them ($I_1$) allows us to take into account the effect of urbanization. The second method ($I_2$) allows us to exclude the effect of urbanization

Invasibility parameter $I_1$ is based on non-standardized values of $N_{alien}$ and $P_{alien}$. So, $N_{alien}$ and $P_{alien}$ variability in this method associated with urbanization was estimated and considered. To do this:

(1) We used the principal component method to create a new variable, the principal component $PC_{N-Palien}$, from the two parameters ($N_{alien}$ and $P_{alien}$); factor loadings for both parameters were 0.958; the eigenvalue of $PC_{N-Palien}$ was 1.834.

(2) We used analysis of variance (ANOVA) to determine the extent to which the $PC_{N-Palien}$ values depended on the community type and the extent to which they depended on the urbanization level and took the $PC_{N-Palien}$ values averaged for different urbanization levels as an invasibility parameter $I_1$.

Invasibility parameter $I_2$ is based on standardized values of $N_{alien}$ and $P_{alien}$. So, $N_{alien}$ and $P_{alien}$ variability associated with urbanization was discarded, i.e., not estimated or considered. To do this:

(1) We used the standardization procedure $m' = (m - m_{avg})/\sigma$ (where $m'$ is the standardized $m$ parameter value, and $m_{avg}$ and $\sigma$ are the average value and standard deviation $m$ parameter in the sample) to calculate the standardized $N_{alien}'$ and $P_{alien}'$ values; standardization was carried out separately for all urban relevés and separately for all non-urban relevés (i.e., $m_{avg}$ and $\sigma$ were calculated separately for 382 urban and 367 non-urban relevés); as a result of this transformation, $N_{alien}$ and $P_{alien}$ variability associated with urbanization was excluded from further analysis.

(2) We averaged the $N_{alien}'$ and $P_{alien}'$ values, obtaining the invasibility parameter $I_2$.

(3) We used ANOVA to determine the extent to which the $I_2$ values depended on a community type.

*2.6. Data Analysis*

To assess the differences between average values of parameters, we used different General Linear Models (GLM), including combinations of only discrete (two-way ANOVA) or combinations of discrete and continual predictors (factors). In two-way ANOVA, we used factors "community/habitat type" ($dF = 9$) and "urbanization" ($dF = 1$). Two-way GLM was used to assess the link between the values of $N_{\text{alien}}$ and $P_{\text{alien}}$. In this case, we used a continual factor ($N_{\text{alien}}$; $dF = 1$) and discrete factor "urbanization" (non-urban or urban areas; $dF = 1$); the dependent variable was $P_{\text{alien}}$. We did not use any random factors in our models. Pairwise differences in the parameters between discrete predictors were estimated using Tukey's HSD test. The link between parameters was additionally estimated using the following: (I) Pearson's correlation coefficient ($r$); the condition for the use of $r$ was the continuity of both parameters; and (II) Spearman's correlation coefficients ($r_S$), which was used if one of the variables changed as ordinal.

Assumptions of the use of GLM (including ANOVA) were investigated by estimating (I) the homogeneity of variances in the compared groups using the Leven criterion and (II) the correlation between the mean parameter values in the groups and their variances. In all cases, the variances were not homogeneous, and there was a positive relationship between the mean parameter values in the groups and their variances. The tried transformations of the original values (logarithm for $N_{\text{alien}}$ and arcsine transformation for $P_{\text{alien}}$) did not fundamentally improve the GLM usage conditions. Therefore, given the large amount of data analyzed, we did not use any transformations of the original parameter values. We did not use FDR corrections for multiple comparisons because the number of factors in the model we used was not large. The calculations were performed using JMP 10.0.0 (SAS Institute Inc., Cary, NC, USA) and STATISTICA 10.0 (StatSoft, Tulsa, OK, USA).

## 3. Results

*3.1. Alien Species Number and Proportion*

In the non-urban areas, alien species are completely absent in bogs (Table 2). In other non-urban community types, the average number of alien species in the relevé varies from 1 to 3. In the city, the average number of alien species is in bogs, 0–1 species in a relevé, and in other community types, 3–7 species in a relevé. In the same communities, $N_{\text{alien}}$ was 1.6–6.2 times higher in the city than in the non-urban areas. Except for bogs, the proportion of alien species in different community types was 3–8% in the non-urban areas and 8–21% in the city. In the same communities, $P_{\text{alien}}$ was 1.7–8.5 times higher in the city than in the non-urban areas. Statistically significant differences between the $N_{\text{alien}}$ and $P_{\text{alien}}$ values in the non-urban areas and in the city were not found for all community types.

There is a strong correlation between the $N_{\text{alien}}$ and $P_{\text{alien}}$ values: in the non-urban areas, $r = 0.936$; in the city, $r = 0.734$; and in both cases, $p \ll 0.0001$. The slope of the curves approximating the dependences between the $N_{\text{alien}}$ and $P_{\text{alien}}$ values is the same in the city and in the non-urban areas (Figure 2). This is evidenced by the absence of a significant interaction between $N_{\text{alien}}$ and urbanization in the two-factor GLM: $N_{\text{alien}}$—$F = 1073.74$, $p \ll 0.0001$; urbanization—$F = 45.17$, $p \ll 0.0001$; and interaction $N_{\text{alien}} \times$ urbanization—$F = 0.34$, $p = 0.5579$. Therefore, both in urban and non-urban areas, a similar increase of $N_{\text{alien}}$ is accompanied by a similar increase of $P_{\text{alien}}$. However, if the average values of $N_{\text{alien}}$ in urban and non-urban areas are similar, the average values of $P_{\text{alien}}$ in urban areas are increased compared with non-urban areas.

*3.2. Invasibility Parameter $I_1$*

The $N_{\text{alien}}$ and $P_{\text{alien}}$ variables were combined into one new invasibility parameter, $PC_{N\text{-}P\text{alien}}$, which explained 91.7% of the variability of the two original variables using the principal component method.

A two-factor ANOVA with "community type" and "urbanization" factors found a strong dependence of $PC_{N\text{-}P\text{alien}}$ on both factors and their interaction: community type ($F = 13.50$, $p \ll 0.0001$); urbanization ($F = 118.94$, $p \ll 0.0001$); and community

type × urbanization interaction ($F = 5.57$, $p << 0.0001$). Therefore, the invasion level, integrally described by the $PC_{N\text{-}P\text{alien}}$ values, firstly, differs between different community types; secondly, differs within a community type between urban and non-urban habitats; and thirdly, the differences between the invasibility of different community types are different in the city and in the non-urban areas.

**Table 2.** Average number ($N_{\text{alien}}$) and proportion ($P_{\text{alien}}$) of alien species in the relevé in different community types in the non-urban areas and in the city (average ± standard error).

| Community Types (EUNIS Habitat Types) | $N_{\text{alien}}$ Non-Urban Habitats | $N_{\text{alien}}$ Urban Habitats | Differences [1] | $P_{\text{alien}}$, % Non-Urban Habitats | $P_{\text{alien}}$, % Urban Habitats | Differences [1] |
|---|---|---|---|---|---|---|
| Fens and bogs without trees (Q1, Q5) | 0 | 0.72 ± 0.23 | n.s. | 0 | 2.8 ± 1.00 | n.s. |
| Swamps, bog woodlands, shrub fens (T16, T3J, S92) | 0 | 0 | n.s. | 0 | 0 | n.s. |
| Flood meadows (R35) | 1.15 ± 0.29 | 3.40 ± 0.74 | n.s. | 3.2 ± 0.8 | 14.9 ± 1.7 | *** |
| Riparian forests (T11) | 1.33 ± 0.39 | 4.19 ± 0.66 | * | 3.4 ± 0.9 | 14.3 ± 2.6 | *** |
| Mesic and wet meadows (R2, R3) | 2.40 ± 0.45 | 4.97 ± 0.59 | ** | 5.6 ± 1.0 | 14.0 ± 1.4 | *** |
| Birch forests (T1) | 1.00 ± 0.22 | 5.50 ± 0.76 | *** | 1.8 ± 0.4 | 15.3 ± 2.1 | *** |
| Coniferous plantations (T359) | 2.77 ± 0.38 | 4.54 ± 0.39 | n.s. | 5.0 ± 0.7 | 8.7 ± 0.7 | n.s. |
| Coniferous forests (T3) | 1.58 ± 0.15 | 4.04 ± 0.15 | *** | 2.6 ± 0.2 | 7.9 ± 0.3 | *** |
| Rocky grasslands (R1A, V3) | 3.31 ± 0.94 | 5.75 ± 1.25 | n.s. | 8.2 ± 2.1 | 14.1 ± 2.8 | n.s. |
| Pine forests on rocky outcrops (T35) | 1.13 ± 0.31 | 7.00 ± 1.00 | * | 2.7 ± 0.6 | 20.5 ± 3.8 | ** |

[1] The significance of differences in Tukey's two-factor ANOVA between non-urban and urban communities; n.s.—not significant; *—$p < 0.05$; **—$p < 0.01$; ***—$p < 0.0001$.

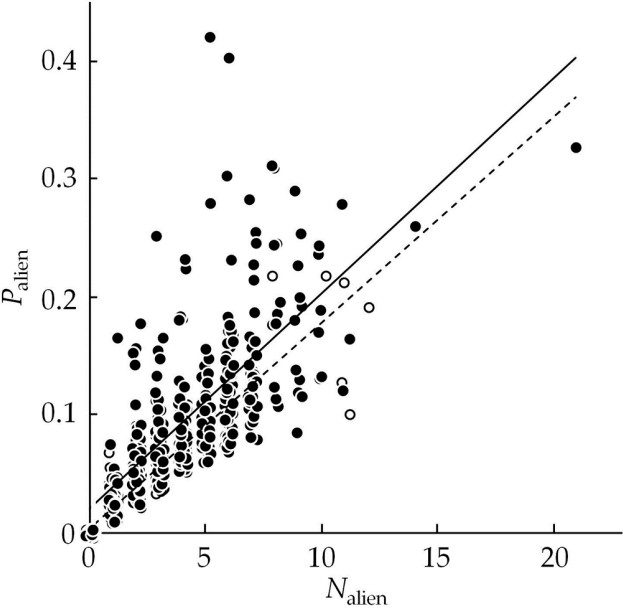

**Figure 2.** Relationship between the number ($N_{\text{alien}}$) and proportion ($P_{\text{alien}}$) of alien species in the relevé in non-urban (○, dotted line) and urban (●, solid line) habitats. A little random noise has been added to the coordinates to reduce point overlap.

By averaging the $PC_{N\text{-}P\text{alien}}$ values obtained for the same community types at different levels of urbanization, we obtained community invasibility $I_1$. In other words, $I_1$ describes the $PC_{N\text{-}P\text{alien}}$ variability component as resulting from variability between different community types (Figure 3). Based on the $I_1$ values, bog communities are the least invasible, while grass meadow communities and treeless areas of rocky grassland on the tops of hills are

the most invasible. The remaining community types, as shown in the structure of pairwise differences estimated using Tukey's test, slightly differ from each other in the average $I_1$ values. However, based on the $I_1$ values, we can discuss or suggest that natural coniferous forests are less invasible than deciduous forests and coniferous plantations.

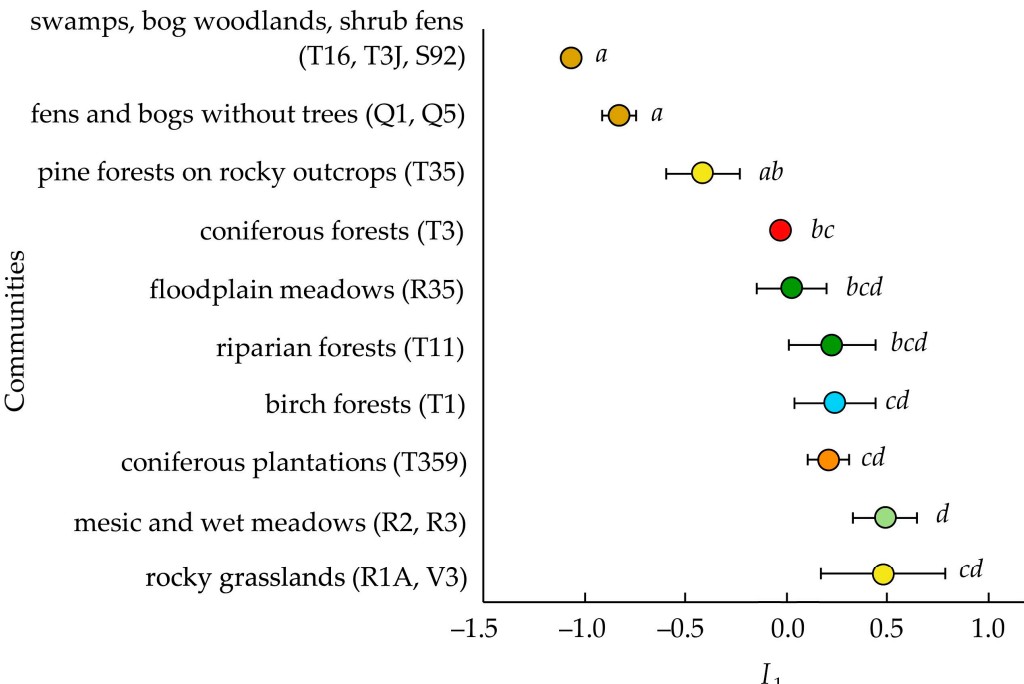

**Figure 3.** Average values of invasibility in all the relevés (dark yellow circles—bog habitats with and without trees; bright yellow circles—rocky habitats with and without trees; red circles—coniferous forests; dark green circles—floodplain habitats with and without trees; blue circles—birch forests; orange circles—coniferous plantations; light green circles—mesic and wet meadows), determined by the first method ($I_1$; ±SE; if the lines are not visible, this means that SE is less than the symbol) in different community types. Letter indices show homogeneous (at $p < 0.05$) values according to Tukey's test.

### 3.3. Invasibility Parameter $I_2$

As for the $I_2$ values determined by the second method, the two-factor ANOVA with "community type" and "urbanization" factors did not find the dependence on urbanization: community type ($F = 12.71$, $p << 0.0001$); urbanization ($F = 1.20$, $p = 0.2737$); and community type $\times$ urbanization interaction ($F = 4.06$, $p < 0.0001$). Consequently, the community invasibility determined by the second method, i.e., $I_2$, differs between different community types but does not consistently differ within a community type between urban and non-urban habitats.

Based on the $I_2$ values, just as based on the $I_1$ values, bog communities are the least invasible, and grass meadow communities and treeless rocky grassland communities on the tops of hills are the most invasible (Figure 4). The other types of communities are homogeneous in terms of $I_2$ values. Invasibility specifics between natural coniferous forests, on the one hand, and deciduous forests and coniferous plantations, on the other hand, are visible only at the trend level, i.e., are not statistically found.

### 3.4. Relationship between $I_1$ and $I_2$

Community invasibility estimates obtained by the two methods correspond to each other in general. The $I_1$ and $I_2$ average values demonstrate a correlation coefficient of $r = 0.94$, $n = 10$; $p < 0.0001$ (Figure 5). At the same time, the slope of the empirical regression line does not differ from the slope of +1.

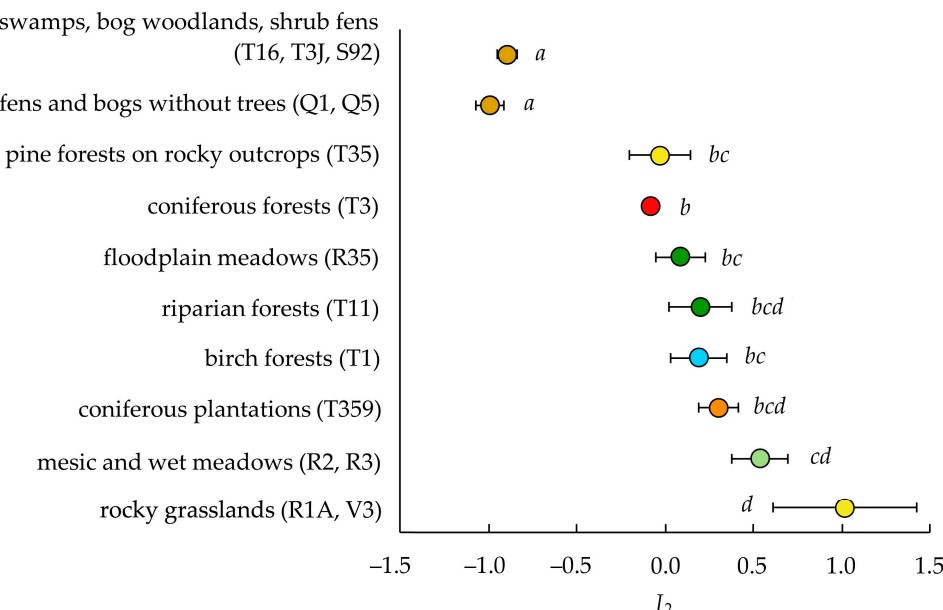

**Figure 4.** Average values of invasibility in all the relevés (dark yellow circles—bog habitats with and without trees; bright yellow circles—rocky habitats with and without trees; red circles—coniferous forests; dark green circles—floodplain habitats with and without trees; blue circles—birch forests; orange circles—coniferous plantations; light green circles—mesic and wet meadows), determined by the second method ($I_2$; ±SE; if the lines are not visible, this means that SE is less than the symbol) in different community types. Letter indices show homogeneous (at $p < 0.05$) values according to Tukey's test.

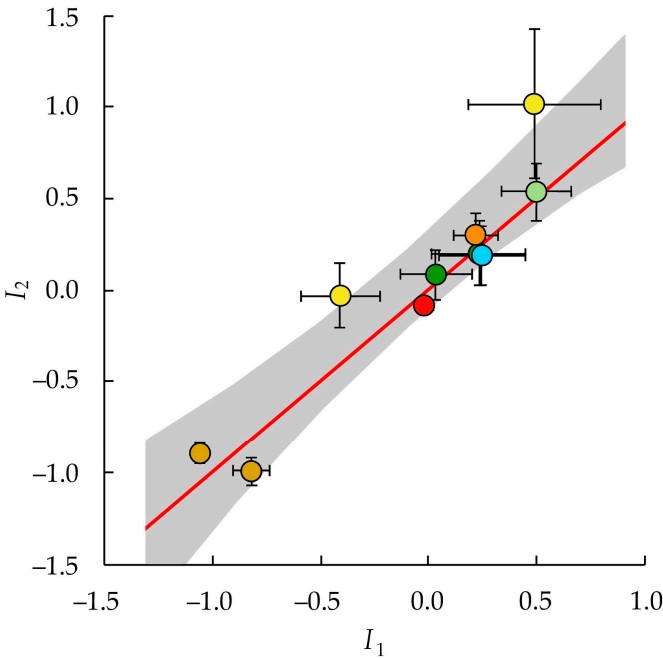

**Figure 5.** Correlation between the average (±SE) $I_1$ and $I_2$ values in 10 types of natural plant communities (dark yellow circles—bog habitats with and without trees; bright yellow circles—rocky habitats with and without trees; red circles—coniferous forests; dark green circles—floodplain habitats with and without trees; blue circles—birch forests; orange circles—coniferous plantations; light green circles—mesic and wet meadows). Gray area shows a 95% confidence interval for the empirical regression line; the red line is a straight line with a slope of +1.

## 4. Discussion

### 4.1. Invasion Levels

We did not analyze the invasion levels and invasibility of human-managed communities, such as yards and parks. This allowed us to select and explore communities of each type at contrasting levels of urbanization. Plots located in the city and in the non-urban areas show different invasion manifestations: in the city, the $N_{alien}$ and $P_{alien}$ values are higher. This is absolutely natural and expected [3,10,22]. It can be explained by the fact that in the city, compared with non-urban areas, the flow of alien species propagules and the degree of community disturbance are higher.

To explain the invasion levels found in urban and non-urban communities, we, first of all, applied the space-time analogy principle. Non-urban communities with low $N_{alien}$ and $P_{alien}$ values are communities with early invasions. The later invasion stages can be seen in urban communities.

Other explanations for the differences between $N_{alien}$ and $P_{alien}$ values between the city and the suburb may be related to (I) different environmental conditions of urban and non-urban habitats; (II) different sources of alien propagules in urban and non-urban habitats; and (III) the effects linked with habitat fragmentation. For example, in the Middle Urals, the levels of invasion in forests showed a significant edge effect [33,34] and were high ($P_{alien}$ = 10–20%) in small fragments of urbanized forest vegetation [35]. However, it was not the task of this study to find the reasons for the differences in the invasion levels in habitats with varying degrees of urbanization. Unexpectedly, the invasion levels in some community types in the Middle Urals and Europe [5] were found to be similar. Reference [5] gives the proportions of neophytes and archaeophytes in different habitats in the Czech Republic and Great Britain. To compare it with our estimates, which were used as $P_{alien}$, we added up the proportions of neophytes and archaeophytes published in [5]. The $P_{alien}$ values for non-urban communities in the Middle Urals (MU) and Europe (E) were found to be close: bogs 0% (MU) and 0.1–0.2% (E); meadows 5.6% (MU) and 3.8–6% (E); deciduous forests 1.8% (MU) and 1.7–4% (E); and coniferous forests 2.7% (MU) and 1–25.1% (E). Considering climate differences, as well as differences in time and intensity of anthropogenic territory development, the Middle Urals would be expected to show lower invasion levels than Europe.

We have not subdivided alien plants into archaeophytes/neophytes or non-invasive/invasive. This division may provide additional information about the invasion processes [2,3,5]. We analyzed alien plants in general because it is believed that focusing solely on currently invasive species may hinder the identification of highly invasible ecosystems [2].

### 4.2. Invasibility Estimation Methods

In accordance with the accepted distinction between "invasion levels" and "invasibility", the invasibility of a community should not depend on its surroundings [3,4]. Both methods used to determine invasibility—$I_1$ and $I_2$—define invasibility as an average level of invasion recorded under different environmental conditions in the studied communities, i.e., both parameters are designed to be independent of the environment.

When calculating $I_1$, independence of the environment appears only at the last stage of calculations when averaging the $PC_{N-P alien}$ values found in the city and in the non-urban areas. When calculating $I_2$, the specifics of invasion levels dependent on urbanization had already been excluded at the first stage of calculations when switching to the standardized $N_{alien}$ and $P_{alien}$ values. Standardization, in fact, was used to level out the specifics of invasion levels resulting from different urbanization manifestations. However, standardization can be done considering other landscape or habitat specifics, such as geographic region, size or degree of habitat disturbance, type and proximity of a propagule community, etc.

The second way to estimate invasibility, $I_2$, seems more suitable for all sorts of comparisons. For example, when comparing community invasibility in two regions using $I_1$,

it is necessary to study the same community types in both regions. When using $I_2$, this observational balance is desirable but not necessary.

The proposed methods for estimating invasibility give estimates on an interval scale because the natural and constant starting points for both $I_1$ and $I_2$ are not determined. At the same time, the $I_1$ and $I_2$ values can be easily converted into ordinal scales by ranking the communities in ascending order of invasibility. $I_1$ and $I_2$ meet 7 of the 13 criteria of a corresponding invasion parameter (invasibility) [2]. $I_1$ and $I_2$ are (1) widely applicable and comparable; (2) not dependent on the scale; (3) measurable; (4) reliable; (5) economical; (6) based on data obtained without destroying ecosystems or biota; and (7) based on available data. The interpretability and unambiguity of $I_1$ and $I_2$ will be assessed when the situations in which they are used accumulate.

Both methods, because they are based on the same initial data, gave similar results and allowed us to arrange the communities from the least to the most invasible (see Figure 5).

### 4.3. Invasibility of Different Community Types

Based on the invasibility degree, we identified three community-type groups with two subgroups in one group. In ascending order of the $I_1$ and $I_2$ values, the communities are arranged as follows: (1) bogs with and without trees; (2a) coniferous forests on high and medium terrain elements; grass communities in floodplains; (2b) deciduous forests, coniferous plantations, floodplain communities with woody plants; and (3) grass communities in mesophytic (meadows) and rocky grassland habitats.

Initially, we assumed that regularly and spontaneously disturbed floodplain communities (as found in [5]), meadows, or artificial plantations might be the most vulnerable to invasions. This assumption was partially confirmed for meadows and plantations. According to our data, the invasibility of floodplain communities in the Middle Urals is not very high, although, in Europe, the proportions of neophytes and archaeophytes in riverine habitats are usually high [5]. The least invasible communities in the Middle Urals are bogs, which, in several studies in Europe [5,21,22], have also been found resistant to the invasion of alien plants. In Australia, however, bogs are habitats with moderate invasibility levels [2].

The high invasibility of grass communities on high and medium terrain elements may be related to the absence of tree dominants. Sometimes vulnerability to alien plants is associated with the absence of competitors [18]. However, not all communities without trees are highly invasible. For example, low or not-high $I_1$ and $I_2$ values were found for bogs and floodplain communities without trees.

The specifics of the invasibility of different community types can be related to the frequency, type, and degree of disturbance and the time since the last one.

In the Middle Urals, grass communities exist only due to constant external disturbances. Mesophytic grass communities—meadows—in the region are only human-made and, without regular mowing, will overgrow with trees in a few decades. On high terrain elements, grass communities are maintained due to unfavorable soil conditions for woody plants: due to strong surface erosion, periodically drying soils are thin. The nature of the *Festuco-Brometea* class rocky steppes in Ekaterinburg and its non-urban areas is successionally unstable and inconsistent with the current climatic conditions, which is confirmed by the recorded overgrowth of many steppe areas with forests [49]. Upland deciduous forests and coniferous plantations are successionally more advanced communities that emerged in the place of cut-down or burned forests or former farmlands. The predominant age of birch in birch forests is 50–90 years, and that of pine in plantations is 40–80 years. These time marks can be considered the time of the last severe disturbances. Pine forests are a regional subclimax. They are represented by post-fire or post-cutting forests with trees aged 80–100 (130) years. This means that at least this much time has passed since the last severe disturbances until the moment of our survey. Bog communities spontaneously developed for the longest time in the Middle Urals. The community type in bogs remains unchanged for at least hundreds of years, while they have existed as habitats for thousands of years [50].

It can be assumed that floodplain communities are also regularly disturbed—less often than grassland and rocky grassland communities but more often than pine forests and bogs.

The time passed since the last severe disturbances that led to the emergence or maintenance of these community types is ranked from I to IV: I—frequently disturbed meadows and rocky grassland communities without trees; II—deciduous forests, plantations, and floodplain communities that emerged after disturbances that took place less than 100 years ago; III—coniferous forests and rocky grassland communities with trees that emerged after disturbances that took place more than 100 years ago; and IV—bogs that have existed for hundreds of years without severe disturbances.

The Spearman's correlation coefficient between the disturbance regularity and the $I_1$ values is $r_S = -0.96$ ($n = 10$, $p < 0.0001$), and for $I_2$, the $r_S$ value is the same (Figure 6).

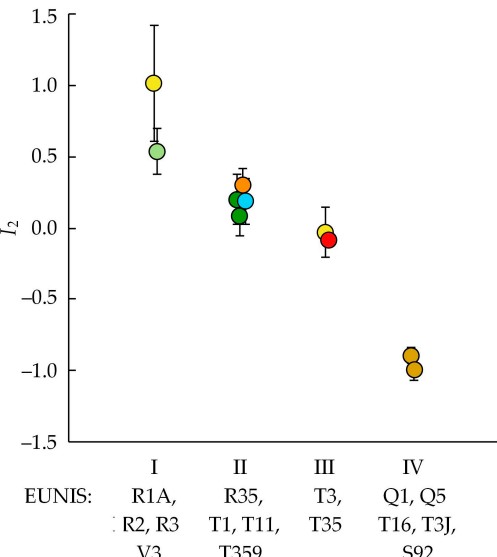

**Figure 6.** Invasibility ($I_2$; ±SE) decreases as the time passed since the last severe community disturbance increases (I–IV). EUNIS community classification codes are indicated. Different habitats are shown by different colors (dark yellow circles—bog habitats with and without trees; bright yellow circles—rocky habitats with and without trees; red circles—coniferous forests; dark green circles—floodplain habitats with and without trees; blue circles—birch forests; orange circles—coniferous plantations; light green circles—mesic and wet meadows).

This means that the less time has passed since the last severe community disturbance, the higher the average invasibility determined for this community type is. Therefore, the frequency, type, and degree of disturbance is one of the possible invasibility factors. The relationship between frequency, type, and degree of disturbance and the spread of alien plants has been discussed many times [1,2,51]. Our estimates confirm the fundamental nature of this relationship.

Other reasons for different invasibility may be related to different amounts of resources and the specifics of the resource availability in different community types [1,4], the specifics of the fragmentation of different community types, such as the typical size of the habitats they occupy, or edge effects [20,33,51]. However, there are currently no empirical or expert estimates of these properties for all community types in the Middle Urals. Therefore, testing the assumptions about their correlation with invasibility remains a task for future research.

## 5. Conclusions

The invasibility of communities can be estimated based on invasion levels—the number and proportion of alien species in the community. At the same time, invasibility estimates are independent of the landscape in which the communities are located and the specifics of neighboring communities. Two methods for estimating invasibility were

proposed, which gave similar results. These invasibility parameters are widely applicable and comparable, scale-independent, measurable, and reliable, based on data that does not require the destruction of ecosystems or biota and are based on available data. Based on the invasibility degree, we identified three community-type groups with two subgroups in one group. In ascending order of invasibility degrees, community types were arranged as follows: (1) bogs with and without trees; (2a) coniferous forests on high and medium terrain elements and grass communities in floodplains; (2b) deciduous forests, coniferous plantations, floodplain communities with woody plants; and (3) grass communities in mesophytic (meadows) and rocky grassland habitats. Obtained results of the assessment of different plant community invasibility may be used to understand patterns of alien plant distribution in local habitats and the reasons for the different vulnerability of communities to plant invasions.

**Author Contributions:** Conceptualization, D.V.V.; methodology, D.V.V. and N.V.Z.; formal analysis, D.V.V. and N.V.Z.; investigation, N.V.Z., D.I.D., E.N.P., L.A.P., and A.A.K.; data curation, E.N.P.; writing—original draft preparation, D.V.V. and A.A.K.; writing—review and editing, D.I.D.; visualization, L.A.P. All authors have read and agreed to the published version of the manuscript.

**Funding:** This research was funded by the Russian Science Foundation, grant number 22-24-20149 (https://www.rscf.ru/project/22-24-20149/, accessed on 25 June 2023). The APC was paid by the author's team.

**Institutional Review Board Statement:** Not applicable.

**Data Availability Statement:** Terms of access to the original data used in the study are discussed upon request to the corresponding author.

**Acknowledgments:** Authors are grateful to T.G. Ivchenko (Komarov Botanical Institute of the Russian Academy of Sciences) for the provided part of the vegetation relevés of the bog communities used in the research. The authors are also grateful to two anonymous reviewers who pointed out errors and inaccuracies in the text and the contents of the manuscript and helped to significantly improve it during the review.

**Conflicts of Interest:** The authors declare no conflict of interest.

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
