# Peer review of "Invasibility of Common Plant Community Types of the Middle Urals"

_diversity, doi:10.3390/d15090955_

Round 1

Author Response

Dear editor,

Dear Reviewers,

we are grateful for friendly and consistent analysis of our manuscript.

We agree, that reviewers’ comments and recommendations helped us to find some unclear or incorrect points, which were in the first version of the manuscript.

Thus, we took into account and corrected most of such points. However, we decided that some of the reviewers’ comments needs not a correction, but a clear description of our opinion on this without corrections, or without significant ones. Here we provide our point-by-point responses to the comments of both reviewers.

With the best regards, 

authors' team

Reviewer 2 Report

The researchers have gathered a large data set from 749 plots located in urban and non-urban habitats to determine the number and proportion of alien species that have invaded the range of vegetation types that occur in the region around Ekaterinburg. They developed two indices based on PC analysis and statistical means and variance, that showed similar results. Urban habitats have more alien species than non-urban, and plant communities that have experienced more recent disturbance also have more alien species.

Studies on alien species sometimes two additional metrics, percent cover of alien species at each site relative to native cover, and size of vegetation patches. Some alien species completely dominate and replace the native vegetation.  The researchers did not collect percent cover or abundance data, but might comment on whether these habitats are completely dominated by alien species, or still have largely native species. The size of vegetation patches determines how much invasion by aliens has occurred, with small patches being more invaded than large patches. I imagine that vegetation patches are smaller in urban than non-urban areas, so these two variables may be confounded and cannot be separated, even if vegetation patch size were known. It may be of value to discuss size of vegetation patches in the Discussion. I have a few questions regarding placement of sample plots in large versus small vegetation patches, see below.

I understand that the goals of this study were limited to number and proportion of alien species, but the authors might address the issues of alien species cover and patch size in the Discussion.

Specific comments follow:

13  The word “suburban” is defined as a residential area that surrounds a city, so the sample design is not apparent from the description in the Abstract. It was not apparent to me that rural areas are included, until I saw the map in Fig. 1. It might be appropriate to use the word “rural” to describe these sites distant from the city. Rural areas include larger expanses of managed or natural vegetation. In fact, you have surveyed a range of habitats from urban to suburban to rural. Perhaps the term “non-urban” (l. 135) is more appropriate, rather than suburban.

14 “the number and proportion of alien species in the description”

I recommend that you clarify this is the proportion of alien species relative to the total number of native species:

the number and proportion of alien species per sample site relative to the number of native species

23 This confirms that the less time has passed

CHANGE TO

This confirms that the less time that has passed

76 I could not find a definition of “edificator” so I do not believe this is a known word. Please use another word, as sentence is unclear.

106 “Invasion levels have been set…”

Do you mean:  Invasion levels have been previously determined….? Can you provide more information about these studies [31-34], either here or in the Discussion?

133 “ vegetation descriptions…”….“The number of descrip-tions differed by more than an order of magnitude between different community types”

By descriptions, do you mean the number of plots, sites or habitats sampled? Is the number of descriptions the same as the number of plots? Please clarify.

136 “Descriptions were made in 2013–2022 on 20x20m plots”

Please be specific about what data were collected in plots. How was proportion of alien species determined? See my comment to line 140.

139 “decade” means 10 years. Do you mean the first 10 days of June to the first 10 days of August? Perhaps this could be simplified to the first week of June to the first week of August.

140 “we recorded all types of vegetative plants found in an area of 100 or 400 m2

Field methods are unclear from this sentence. Is this what you did: we recorded all plant species found in an area of 100 or 400 m2

Table 1. What criteria did you use to differentiate suburban and urban habitats? Density of houses or other development? As stated above, I suggest you use rural or non-urban habitats rather than suburban.

What about size of the sampled habitats? Larger vegetation patches may have less invasion in the center than the edges. Were sample plots placed on the edges of large vegetation patch habitats, (e.g., near settlements or roads), or in the center?

185 “average number of alien species in the description”

The use of the word “description” is unclear here, too. Does this mean: average number of alien species per sample site? Please clarify. “Description” is used throughout the manuscript.  I suggest you use sample plot, site or habitat, or edit otherwise for clarity.

284-285 What is BC? Please describe and spell out.

363 “It is difficult to introduce floodplain communities into the described range,,,,”

Do you mean:  It is difficult to estimate disturbance intervals in floodplain communities….? Please clarify.

It seems to me that Figure 6 should be first reported in the Results. Discussion material related to Fig. 6 should, of course, remain in the Discussion. However, I leave this up to the editor.

Overall, the manuscript is well-written and the English is good, except for a few places where I have noted unclear sentences or unclear word choice.

Author Response

(The authors gave the same response as above.)
